# Four Invasive Plant Species in Southwest Saudi Arabia Have Variable Effects on Soil Dynamics

**DOI:** 10.3390/plants12061231

**Published:** 2023-03-08

**Authors:** Ahmed M. Abbas, Wagdi S. Soliman, Maryam M. Alomran, Nahaa M. Alotaibi, Stephen J. Novak

**Affiliations:** 1Department of Biology, College of Science, King Khalid University, P.O. Box 9004, Abha 61413, Saudi Arabia; 2Department of Botany and Microbiology, Faculty of Science, South Valley University, Qena 83523, Egypt; 3Horticulture Department, Faculty of Agriculture and Natural Resources, Aswan University, Aswan 81528, Egypt; 4Department of Biology, College of Science, Princess Nourah bint Abdulrahman University, P.O. Box 84428, Riyadh 11671, Saudi Arabia; mmalomran@pnu.edu.sa (M.M.A.); namialotaibi@pnu.edu.sa (N.M.A.); 5Department of Biological Sciences, Boise State University, Boise, ID 83725, USA

**Keywords:** *Prosopis juliflora*, *Ipomoea carnea*, *Leucaena leucocephala*, *Opuntia ficus-indica*, soil properties, soil microelements, invasive plant species, ecosystem modification

## Abstract

Predicting the direction and magnitude of change in soil dynamics caused by invasive plant species has proven to be difficult because these changes are often reported to be species- and habitat-specific. This study was conducted to determine changes in three soil properties, eight soil ions, and seven soil microelements under established stands of four invasive plants, *Prosopis juliflora*, *Ipomoea carnea*, *Leucaena leucocephala*, and *Opuntia ficus-indica*. Soil properties, ions, and microelements were measured in sites invaded by these four species in southwest Saudi Arabia, and these values were compared to the results for the same 18 parameters from adjacent sites with native vegetation. Because this study was conducted in an arid ecosystem, we predict that these four invasive plants will significantly alter the soil properties, ions, and microelements in the areas they invaded. While the soils of sites with the four invasive plant species generally had higher values for soil properties and ions compared to sites with native vegetation, in most instances these differences were not statistically significant. However, the soils within sites invaded by *I. carnea*, *L. leucocephala*, and *P. juliflora* had statistically significant differences for some soil parameters. For sites invaded by *O. puntia ficus-indica*, no soil properties, ions, or microelements were significantly different compared to adjacent sites with native vegetation. Sites invaded by the four plant species generally exhibited differences in the 11 soil properties, but in no instance were these differences statistically significant. All three soil properties and one soil ion (Ca) were significantly different across the four stands of native vegetation. For the seven soil microelements, significantly different values were detected for Co and Ni, but only among stands of the four invasive plant species. These results indicate that the four invasive plant species altered soil properties, ions, and microelements, but for most of the parameters we assessed, not significantly. Our results do not support our initial prediction, but are in general agreement with previous published findings, which indicate that the effects of invasive plants on soil dynamics vary idiosyncratically among invasive species and among invaded habitats.

## 1. Introduction

In regions where native plants and animals did not meet the needs of early colonists, these colonists began moving plants and animals around the globe, with some of these non-native species becoming invasive [1,2]. Invasive plant species are often described as having negative impacts on ecosystem processes [3]. They not only pose a threat to global biodiversity hotspots, but also to all the habitats (i.e., forest, grasslands, etc.) they invade. There are many ways that invasive species damage native communities and ecosystems, especially in remote locations and on islands, where native species may be more vulnerable to these non-natives [4,5]. The extent to which invasive plants alter ecosystem processes in their new range compared to the ecosystem processes of native species in these areas has been shown to vary among invasive species and invaded habitats [6,7,8]. Many accounts of the impacts of invasive species are available for certain habitats and regions around the world [9,10], but details on the impacts of invasive species in other habitats and regions (e.g., in Saudi Arabia) are largely unknown, but see Assaeed et al. [11].

Invasive plant species can cause rapid changes in soil properties relative to native communities, and these differences may hinder efforts to restore native plant communities [8,12]. Invasive plants modify soil conditions either directly, by depositing leaf litter of varying quality and quantity [13,14], or indirectly, by altering the native microbial communities and the biotic interactions they normally engage in [14,15]. Invasive plants may alter soil properties in ways that increase their competitive advantage. For example, the soils of sites infested with invasive plants have lower levels of accessible phosphorus, total nitrogen, and carbon than sites with native vegetation [16,17]. However, the soil carbon/nitrogen ratio of soils under invasive plants has been reported to increase [8]. Additionally, the capacity of soils in invaded wetlands to store and release carbon was influenced by plant biomass input in soils, the sediment deposition rate of the tidal salt marsh, carbon turnover, and organic carbon stability, which are generally reported to be associated with the accumulation and decomposition of organic matter [18]. Invasive species in tropical dry forests negatively alter soil characteristics [19]. In contrast, the presence of invasive species in degraded habitats can improve soil moisture conditions and can facilitate the reclamation of such degraded areas [20].

Invasive species that displace native plants create unique plant communities that alter or create new plant–soil nutrient cycles and carbon cycles and change the capacity of soil to accumulate N and P [21]. In fact, the ability of invasive plants to alter the concentration and stoichiometry of soil nutrients has been widely studied [22,23,24,25]. Some invasive plants may alter the soil properties (and other ecological processes) to such an extent that the ecosystem services associated with native communities are threatened or are no longer provided [19,26,27,28]. Such invasive plant species are referred to as “transformer species” [29]. However, recent assessments of invasive plant species highlight the need to distinguish between non-native (invasive) species that have minimal ecological consequences and non-native species that have major ecological consequences in their new habitats [30,31]. There is ample evidence that invasive species can alter ecological processes and reduce native biodiversity, yet the ecological consequences of invasive tree species and how they modify soil properties and conditions are still not well understood and not adequately documented [32].

Other studies have reported no statistically significant difference among the properties and microelements of soils from habitats infested with invasive plant species compared to the soils of equivalent habitats occupied by native plant species [6,8,12,32,33]. Moreover, several studies that have compared the capacity of multiple invasive species to alter soil properties within the same experiments design have revealed contrasting results among the species examined [6,8,32]. Other studies have reported that the same invasive species alters soil dynamics in different ways in different habitats (communities) or under different experimental conditions [6,8,12]. These results indicate that the effects of invasive plants on soil dynamics vary in an idiosyncratic manner among invasive species and among invaded habitats (see [6,8]). Thus, the alteration of soil properties by invasive plants is often species- and habitat--specific.

Invasive plant species in Saudi Arabia have negative ecological consequences and high economic costs in natural ecosystems and agriculture areas, respectively [34]. These invasive plants have displaced many native plants from species-rich areas, especially in high-elevation habitats, wadis, and meadows. Some non-native plant species are believed to have been introduced into Saudi Arabia during past Hajj seasons, while others were introduced by early settlers for use as food, medicine, ornamental uses, livestock consumption, shade, and other reasons [34]. The introduction of non-native plants continues and is even increasing due to increased international trade, commerce, travel, and accidental means [35]. In Saudi Arabia, the establishment of non-native plant species is also facilitated by human disturbance of natural habitats, including the removal of native plants for road-building activities. This process explains the presence of invasive species along roads in mountainous regions of southwest Saudi Arabia [34]. Additionally, many other regions of southwest Saudi Arabia, plant diversity hotspots such as wadis and Red Sea islands, now contain many invasive plant species [34]. Unfortunately, scientific research focused on the invasive plant species in this region of Saudi Arabia is lacking (but see Assaeed et al., [11] and Alfarhan et al., [36]). Thus, there is an urgent need to conduct research on the invasive plants in southwest Saudi Arabia and to better understand their ecological consequences.

Six non-native plant species have been identified as the most important invasive species in Saudi Arabia [34]: *Prosopis juliflora*, *Cuscuta campestris*, *Cenchrus echinatus*, *Verbesina encelioides*, *Leucaena leucocephala*, and *Prosopis koelziana*. Among these six species, *P. juliflora* is the most damaging invasive plant species. It was reported as a ruderal plant, mostly found in urban and suburban areas, and is rarely observed in native plant communities of the central region of Saudi Arabia [34]. Unlike other tropical and subtropical countries, the invasion of *L. leucocephala* in Saudi Arabia does not have severe impacts on native communities because it mostly occupies urban areas. *Opuntia ficus-indica* and *Opuntia dillenii* are widely distributed across Saudi Arabia. *Ipomoea carnea* ssp. *fistulosa*, a pantropical invasive plant, has wide ecological amplitude. According to Al-Sodany [37], the growth of *I. carnea* follows a seasonal pattern in which its growth is greatest from September and October. This study reported significant differences for growth parameters (e.g., height, crown diameter, crown volume, and size index), demographic characteristics (e.g., birth, mortality, and survival rates), and reproductive output (e.g., number of flowers and fruits) among populations of *I*. *carnea* in different habitats [37]. *I. carnea* subsp. *fistulosa* reproduces clonally, through the production of underground rhizomes that grow laterally through the soil [38].

This research was designed to assess the direction and magnitude of change in soil properties, soil ions, and soil microelements caused by the invasion of four invasive plant species, *Prosopis juliflora*, *Ipomoea carnea*, *Leucaena leucocephala*, and *Opuntia ficus-indica*, in southwest Saudi Arabia. This was achieved by sampling and analyzing soils in areas invaded by the four species and comparing these results to those obtained for soils from adjacent areas where native vegetation occurred. A meta-analysis by Waring et al. [39] showed that the ability of plants to alter soil dynamics is significantly stronger in arid ecosystems; therefore, we predict that all four invasive plants included in this study will significantly alter soil dynamics in the areas they have invaded. The results of this study will be used to test this prediction.

## 2. Results

### 2.1. Comparison between Invaded and Native Soil Samples

The soil properties, ion concentrations, and microelement concentrations of sites in southwest Saudi Arabia invaded by *Prosopis juliflora*, *Ipomoea carnea*, *Leucaena leucocephala*, and *Opuntia ficus-indica* were compared to the soils from sites with native vegetation. Most soil properties, ion concentrations, and soil microelements concentrations did not significantly differ between the *P. juliflora* stands and native stands, except for several microelements, including Cr, Co, Cu, and Ni (Table 1). The concentration of these four microelements was significantly higher for the soil under native vegetation compared to the soil of sites invaded by the tree *P. juliflora*. Except for pH, organic matter (OM), Ca, and SO_4_, no soil properties, ions, or microelements were significantly different for soils from stands invaded by the shrub *I. carnea* and soils from sites with native vegetation (Table 2).

The pH was significantly higher for the soil under native vegetation, while OM, CA, and SO_4_ were significantly higher for the soils from stands with an *I. carnea* overstory (Table 2). Soil properties, ion concentrations, and soil microelements did not significantly differ between stands in which the tree *L. leucocephala* had invaded and stands with native vegetation, except for pH, which was significantly higher in soil from areas with native vegetation (Table 3). No soil properties, ion concentrations, or soil microelements were significantly different for soils from stands invaded by the cactus *O. ficus-indica* and stands of native vegetation (Table 4).

### 2.2. Soil Properties, Ion Concentrations, and Microelement Concentrations for the Four Invasive Plant Species and Adjacent Stands of Native Vegetation

There were large differences in soil properties and ion concentrations in the soils from stands with native vegetation and the soil of stands invaded by the four invasive plant species. We detected significant differences in soil pH, EC, OM, and Ca among the four stands of native vegetation, but not for the stands of the four invasive species (Figure 1). In soils from the native sites, pH ranged from 7.1 to 7.7, and in the invaded site pH ranged from 6.95 to 7.35. pH values were higher in the areas invaded by *P. juliflora* and *O. ficus-indica* (compared to adjacent native sites) and lower in the areas invaded by *I. carnea* and *L. leucocephala* (compared to adjacent native sites). Conversely, EC and OM values were higher in the soils of invaded stands compared to the soil of native sites. The largest increase in the values of EC and OM occurred for sites invaded by *L. leucocephala* (Figure 1). Similarly, the values of Na, Ca, and Mg were highest in the soils of stands that were invaded by all four invasive plants, and the largest values were observed in areas invaded by *L*. *leucocephala* (Figure 1). The concentrations of K in soils of invaded stands of all four invasive species and adjacent native stands were not significantly different (Figure 1). Variation among soil SO_4_ values for the four invasive species in both invaded and native stands were not significantly different (Figure 1). However, the NO_3_ and ammonia values were generally higher in the soils of stands invaded by the four species compared to the soils of sites with native vegetation. The largest increases in NO_3_ values were observed for areas invaded by *P. juliflora*, followed by areas invaded by *L. leucocephala*, with the largest increases in ammonia values occurring in areas invaded with *I. carnea*, followed by areas infested with *P. juliflora* (Figure 1). The soil PO_4_ content exhibited variable results (Figure 1). For instance, the soil PO_4_ content increased in the stands invaded by *O. ficus-indica* compared to the soil of native stands and decreased in areas invaded by *P. juliflora* (Figure 1).

There were also differences in the concentration of microelements in the soils from stands invaded by the four invasive plant species and adjacent stands of native vegetation (Figure 2). The concentration of Mn, Zn, and Fe showed higher values in areas invaded by *O. ficus-indica* and *L. leucocephala*, while these microelements had lower concentrations in areas invaded by *P. juliflora* and *I. carnea*. These differences, however, were not statistically significant. In addition, variation in Cu and Cr in the four stands of native vegetation and the four sites with invasive plants were not significantly different, but values for both microelements were lowest for the soil of sites invaded by *P*. *juliflora* (Figure 2). The concentrations of Co and Ni significantly differed among stands of the four invasive plant species and was significantly lower for the soils of sites invaded by *P. juliflora*, *I. carnea*, and *L. leucocephala*. The four stands of native vegetation exhibited differences for all seven microelements, but none of these differences were statistically significant (Figure 2).

## 3. Discussion

Invasive species are those species (taxa) that are introduced outside of their native range and persist, reproduce, and spread beyond their original points of introduction in their new range [40,41]. Invasive plant species can outcompete and reduce the abundance of native plant species [42], alter soil properties [26], and homogenize the biodiversity of invaded communities and ecosystems [43]. Invasive plant species may alter soil properties in their new ranges, but the magnitude and direction of these changes are sometimes unique to each invasive species and often vary among habitats for the same non-native species [6,8,12,32,33,44,45]. To the best of our knowledge, this is the first study to investigate the biogeochemical consequences of the invasion of *Prosopis juliflora*, *Ipomoea carnea*, *Leucaena leucocephala*, and *Opuntia ficus-indica* in Saudi Arabia. At the outset of this study we predicted that all four invasive plant species would significantly alter soil properties and microelements in the areas they invaded. Our results do not support this prediction.

Soil carbon, nitrogen, and phosphorus have a large influence on many ecosystem processes, and the direction of change for these three elements was widespread across the species and sites studied. Our study revealed higher levels for soil carbon, nitrogen, and phosphorus levels in soils of invaded habitats compared to the soils of sites with native vegetation. A recent meta-analysis discovered that invasive plant species were associated with significantly higher litter deposition and decomposition rates as well as increases in soil nitrogen mineralization and nitrification [46,47]. Ehrenfeld [48] and Ehrenfeld et al. [13] proposed that the characteristics of many invasive plant species, in comparison to native plants, may influence the specific way in which invasive plants alter ecosystem properties. These characteristics include larger size, faster growth, higher photosynthetic rates, higher nutrient concentrations in living tissue and litter, and extremely efficient nitrogen-fixing symbioses. Rout and Callaway [49] state that if invasive species increase net primary production and nitrogen cycling in their invasive ranges but not in their native ranges, their inherent traits are unlikely to drive these processes because these same ecosystem effects should also occur in their native ranges. This statement also highlights the context-specific nature of traits on invasion outcomes.

Another possibility is that invasive plants, once established, will undergo rapid evolution through natural selection [50,51], especially for key traits related to nutrient cycling. For example, Feng et al. [47] provided evidence that the invasive plant *Ageratina adenophora*, which is native to Mexico but has spread throughout subtropical regions around the globe, appears to have developed leaves that decompose more easily in its invaded range. Determining whether and how invasive species have more negative ecosystem consequences than native species can answer critical questions in ecology, evolution, and biogeography [52].

We detected higher NO_3_ levels in the soil from sites infested with all four invasive species compared to soils with native vegetation, even though these differences were not statistically significant. This trend is consistent with previous research indicating a positive relationship between the levels of soil NO_3_ and the abundance of invasive plant species [53]. Previous studies have also reported that plant invasions occur due to several mechanisms that help them uptake and use nutrients more efficiently [23,54]. The mechanisms associated with invasiveness appear to be different in nutrient-poor soils compared to nutrient-rich soils. Many studies indicate that the success of invasive plants in nutrient-poor soils is dependent on strategies such as increased nutrient-use efficiency [24,55], especially at shorter time scales [22]; long nutrient residence times [56]; high resistance to low levels of nutrients [57]; and high plasticity of stoichiometric ratios [23].

We also found that the soil concentrations of Na, Ca^2+^, and Mg^2+^ increase in the soils of invaded areas compared to soils from areas with native plants. According to Duda et al. [58], as *Halogeton glomeratus* invades native *Krascheninnikovia lanata*-dominated vegetation, soil concentrations of nitrate, phosphorus, potassium, and sodium increases with invasion. Rodgers et al. [59] reported that the availability of nitrogen, phosphorus, calcium, and magnesium in the soil, as well as the soil pH, were consistently and considerably higher in soils in North American temperate deciduous forests invaded by the European forb *Alliaria petiolata*. Similarly, our results reveal higher values for most soil properties and soil ions in sites invaded by the four invasive plant species included in this study, even if many of these values were not statistically significant (Table 1, Table 2, Table 3 and Table 4, Figure 1).

Invasive plants frequently play a role in modifying microelement (heavy metal) biogeochemical cycles in ecosystems [60]. Invading plants can increase soil microelement bioavailability in different ways: microelement secretion from leaves and release via humus breakdown and desorption. For instance, most rooted macrophytes can operate as a nutrient pump to absorb the nutrients contained in sediment pools, such as N and P, and subsequently remove them by tissue leaching and secretion and litter decomposition above the soil surface [61]. Invasive plants that perform a similar function can increase the bioavailability of soil microelements [62]. Additionally, microelements contained in plant tissues can be released into the environment by degradation of litter, other organic matter, and diagenesis [62,63]. During the decomposition of invasive plant litter, an increase in soil pH might indirectly result in the mobilization of microelements in the soil. Microelements can be released as sulfides from insoluble sulfide compounds in sediments. The microelements are then oxidized by sediment oxidation and enzymes in the roots of invasive plants [62]. In the same way, plant growth can cause microelements to be released from sediment particles into the pore water of wetlands and concentrate microelements in the rhizosphere of plants [64,65]. This can also be carried out by some native plants.

Increased mobilization can lead to higher rates of leaching and faster uptake of these microelements by plants, which can lower their total concentration in soil by a large amount [66], even though this would be expected to increase bioavailability. In addition to their role in the biogeochemical cycling of microelements, some invasive plants can reduce microelement mobility by adsorption, absorption, immobilization, and “returning elements to belowground pools”. *Spartina alterniflora*, the most pervasive invasive plant in China’s coastal wetland, has caused a considerable decrease in Cs^137^ and Pb^210^ adsorption from beach deposition of silt–fine sand and mud–sand [60]. The roots of hydrophytes, especially invasive hydrophytes, frequently show signs of iron plaque. As a major mechanism for tolerating hazardous metals, it adsorbs and immobilizes heavy metals efficiently, hence reducing their mobility [67]. Furthermore, invasive plants with interactions with symbiotic microorganisms may be able to sequester microelement toxicity through microbial biosynthesis of antioxidants and enzymes [68]. Therefore, the appearance of invasive species in areas contaminated with toxic microelements may not increase the potential enrichment of these toxic elements in the food chain and would in fact contribute to phytostabilization [69].

The results of this study reveal that the microelement values of soils sampled from three of the invasive species analyzed are generally lower than that of the soils from adjacent stands of native vegetation (Table 1, Table 2 and Table 3). The only exception to this pattern occurred for soils from sites invaded by *O*. *ficus*-*indica* (Table 4). Among the four invasive plant species, there were no significant differences in the levels of Mn, Zn, Fe, Cr, and Cu in the soils where they invaded; however, the levels of Co and Ni were significantly lower for soils invaded by *P*. *jliflora*, *I. carnea*, and *L*. *leucocephala* compared to those invaded by *O*. *ficus*-*indica* (Figure 2).

## 4. Materials and Methods

Saudi Arabia contains a variety of ecosystems, ranging from high-altitude mountains (Jabals) (up to 3050 m asl), valleys or river drainages that are seasonally dry (wadis), meadows (Raudhas), salt pans (Sabkhas), lava flows (Harrats), deep sands (Nafud), and irrigation canals. Winters are generally cool with occasional frost in the mountains of the northwestern region. The summer months are hot, with temperatures sometimes reaching above 50 °C. Rainfall is typically unpredictable and erratic in most parts of the country and mainly falls in winter and spring (100–150 mm), except for the southwestern region of the country, which can receive much more precipitation (>600 mm from September to November). Humidity levels are low in the interior of the country (15–20%) and high along the coast (55–75%).

### 4.1. Species Selection

Four species were selected for study: *Prosopis juliflora*, *Ipomoea carnea*, *Leucaena leucocephala* and *Opuntia ficus-indica*. All these species form mono-specific stands in invaded localities and are recognized as being among the most invasive plant species in Saudi Arabia [34].

### 4.2. Field Sampling

Within each habitat, large stands of the four invasive plant species were selected to serve as “invaded” sampling locations. The size of the mono-specific stands varied and was usually 10–12 ha for *P. juliflora*, 2–6 ha for *I. carnea*, 0.6–1 ha for *L. leucocephala*, and 0.5–2 ha for *O. ficus-indica*. The invaded study sites were heavily infested by one of the four invasive species and were located on one side of a distinct ecotone. On the other side of the ecotone were areas consisting of native vegetation; these were the “native” sampling locations. Our experimental design therefore consisted of paired invaded and native sampling locations for each of the four invasive species. Fieldwork was conducted on 3 March 2022 for *P. juliflora*, 15 March 2022 for *I. carnea*, 3 April 2022 for *L. leucocephala*, and 23 April 2022 for *O. ficus-indica*.

Soil samples from the paired invaded and native sampling locations were generally located 20 m from each other. Soil samples were collected at a depth of 0–15 cm at each sampling location. At each invasive and native sampling location, a 20 m transect line was positioned perpendicular to the axis of the ecotone. Ten subsamples were collected at 2 m intervals along the transects established in invaded and native stands at each sampling location and these 10 subsamples were combined into one homogenized soil sample. Six sampling locations were selected for collecting paired soil samples for each of the four invasive plant species, yielding a total of six replicates per species. This approach means that we collected 120 soil subsamples for each species, and a total of 480 soil subsamples for the entire study.

### 4.3. Soil Analyses

Each soil sample replicate was separated into two subsamples. One subsample was dried at 105 °C for standard chemical assays. The soil pH and electrical conductivity (EC) were measured electrometrically in two different soil–H_2_O solutions (1:1, vol/vol and 1:2, vol/vol, respectively) [70]. The total carbon and nitrogen content were determined using dry combustion (FLASH EA1112 CHN analyzer; Thermo Fisher, Waltham, MA, USA). The available P was determined with the Olsen method [71]. Extractable microelements were quantified following digestion with diethylene-triamine-penta-acetic acid [72]. We determined the concentration of seven microelements: manganese (Mn), zinc (Zn), iron (Fe), chromium (Cr), cobalt (Co), copper (Cu), and nickel (Ni). The total microelement content was quantified following digestion with nitric acid [73]. The analysis was conducted using inductively coupled plasma optical emission spectrometry (Spectro Genesis; SPECTRO Analytical Instruments, Kleve, Germany). The second soil subsample was used to determine nitrate and ammonium colorimetrically, following 2 M potassium chloride extraction [74]. Nitrate and ammonium were quantified at the end of the incubation period.

### 4.4. Statistical Analysis

The data generated in this study were tested for normality using the Kolmogorov–Smirnov test and for homogeneity of variance using the Brown–Forsythe test. For each of the four invasive plant species, mean values were calculated based on the results of the six replicates for each species and variation in the data was calculated as the standard error (SE) of the mean. Paired *t*-tests were used to determine differences between invaded and native soil samples for each of the four invasive species. Statistically significant differences in the soil properties, ion concentrations, and microelement concentrations for the stands of the four invasive plants and the native stands were tested using one-way analysis of variance (ANOVA, *F* test) [75]. The Tukey’s honestly significant difference (HSD) test was used to determine the significance among means for treatments with a significant *F*-test. A significance level of *p* < 0.05 was applied for all analyses. All statistical analyses were performed using JMP (version 4.0 SAS Institute, Cary, NC, USA).

## 5. Conclusions

The results reported here do not support our prediction that we would detect significant changes in soil properties and microelements in areas infested by the four invasive plant species compared to the soils from stands of native vegetation. However, our results do indicate that the alteration of soil parameters varies among the four invasive plant species and varies for the different soil parameters we examined. For instance, the soils of sites with the four invasive plant species generally had higher values for soil properties compared to sites with native vegetation, but in most cases these differences were not statistically significant. The soils of stands invaded by *P*. *juliflora* and *I*. *carnea* had the largest number of statistically significant differences: four microelements and four soil properties, respectively. While the soils of sites invaded by *O. ficus*-*indica* and sites with native vegetation exhibited no significant differences for the 18 soil parameters. The soil of areas invaded by the four invasive species showed no significant differences for the three soil properties and the eight soil ions we measured, and no significant differences for five of the seven soil microelements. The levels of Co and Ni were significantly higher for the sites invaded by *O. ficus-indica* compared to the levels of the other three invasive species. Our results are in general agreement with previously published findings that indicate that the effects of invasive plants on soil dynamics vary idiosyncratically among invasive species and among invaded habitats (i.e., these effects are species- and habitat specific). These four invasive plants may simply be incapable of significantly altering soil properties and soil microelements. Conversely, our results do suggest a trend that might signal significant alteration of soil dynamics in the future, especially if the abundance of these invasive plant species increases. Thus, the invasion of these four plant species in southwest Saudi Arabia should be closely monitored and studied to determine their ecological consequences and their impacts on the native vegetation in this region.

## Figures and Tables

**Figure 1 plants-12-01231-f001:**
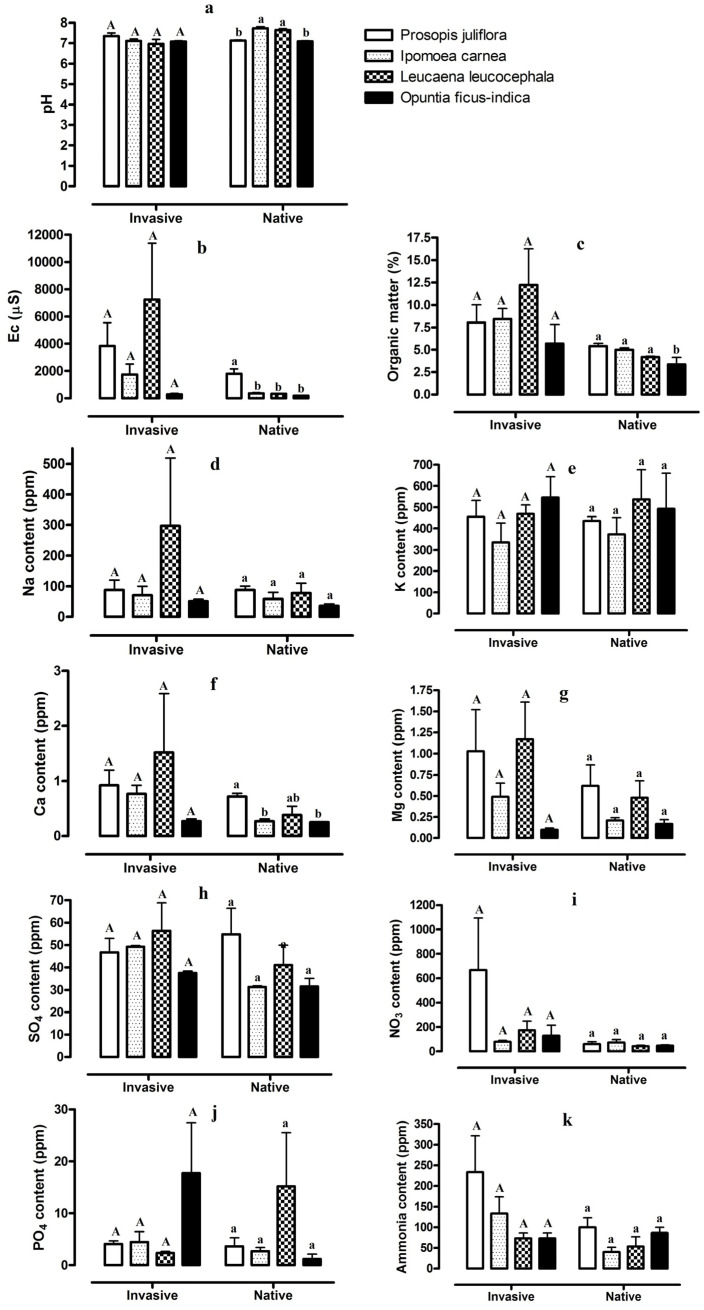
Values of soil properties and ion concentrations, pH (**a**), EC (**b**), organic matter [OM] (**c**), Na (**d**), K (**e**), Ca (**f**), Mg (**g**), SO_4_ (**h**), NO_3_ (**i**), PO_4_ (**j**), and ammonia (**k**), from stands invaded by *Prosopis juliflora*, *Ipomoea carnea*, *Leucaena leucocephala*, and *Opuntia ficus-indica* and adjacent stands of native vegetation in southwest Saudi Arabia. Different capital letters indicate significant differences among the soils from the stands invaded by four invasive plant species and different lower-case letters indicate significant differences among the soils from stands of native vegetation (*n* = 6, *p* < 0.05).

**Figure 2 plants-12-01231-f002:**
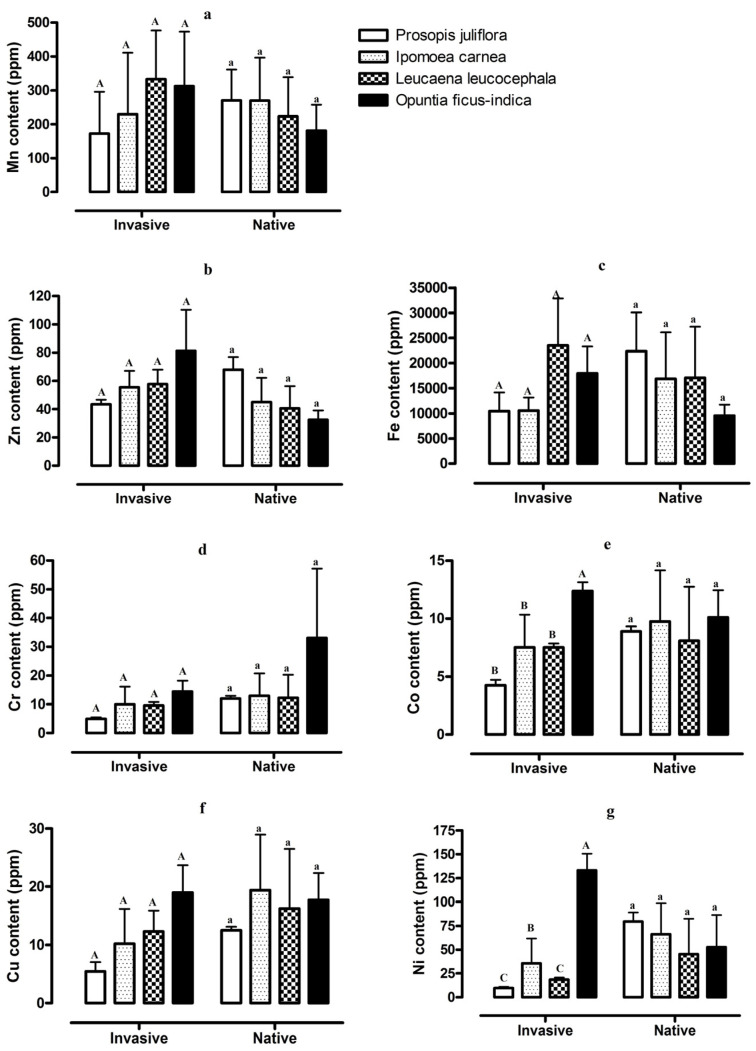
Values of soil microelements, Mn (**a**), Zn (**b**), Fe (**c**), Cr (**d**), Co (**e**), Cu (**f**), and Ni (**g**), from stands invaded by *Prosopis juliflora*, *Ipomoea carnea*, *Leucaena leucocephala*, and *Opuntia ficus-indica* and adjacent stands of native vegetation in southwest Saudi Arabia. Different capital letters indicate significant differences among the soils from stands invaded by four invasive plant species and different lower-case letters indicate significant differences among the soils from stands of native vegetation (*n* = 6, *p* < 0.05).

**Table 1 plants-12-01231-t001:** Soil properties, ion concentrations, and microelement concentrations of soils in stands invaded by *Prosopis juliflora* and adjacent stands with native vegetation in southwest Saudi Arabia.

	Invasive	Native	*t*-Test	*p*-Values
Soil properties				
pH	7.35 ± 0.15	7.13 ± 0.03	1.960	0.234
EC μS	3837.67 ± 1712.79	1797.67 ± 357.34	1.359	0.308
OM %	8.03 ± 1.99	5.39 ± 0.31	1.726	0.259
Ions (cations and anions)				
Na⁺ ppm	88.09 ± 33.55	87.95 ± 12.63	0.001	0.982
K⁺ ppm	454.52 ± 77.13	435.58 ± 20.53	0.056	0.824
Ca^2+^ ppm	0.92 ± 0.28	0.72 ± 0.06	0.477	0.528
Mg^2+^ ppm	1.03 ± 0.49	0.62 ± 0.25	0.551	0.499
SO_4_^2−^ ppm	46.68 ± 6.28	54.89 ± 11.58	0.389	0.567
NO_3_^−^ ppm	666.67 ± 425.57	60.00 ± 20.00	2.028	0.228
PO_4_^3−^ ppm	4.08 ± 0.60	3.59 ± 1.66	0.078	0.794
Ammonia (NH_4_^+^) ppm	233.33 ± 88.19	100.00 ± 23.09	2.139	0.217
Microelements				
Mn ppm	172.92 ± 122.92	270.83 ± 90.60	0.411	0.556
Zn mg/L	43.54 ± 3.15	67.92 ± 8.90	6.665	0.061
Fe ppm	10,445.83 ± 3732.17	22,385.42 ± 7700.57	1.947	0.235
Cr ppm	4.99 ± 0.47	12.05 ± 0.90	48.472	0.002
Co ppm	4.25 ± 0.48	8.91 ± 0.42	53.902	0.002
Cu ppm	5.42 ± 1.63	12.50 ± 0.63	16.514	0.015
Ni ppm	9.69 ± 1.31	79.58 ± 9.42	54.015	0.002

Data represented as means ± SE, *n* = 6.

**Table 2 plants-12-01231-t002:** Soil properties, ion concentrations, and microelement concentrations of soils in stands invaded by *Ipomoea carnea* and adjacent stands with native vegetation in southwest Saudi Arabia.

	Invasive	Native	*t*-Test	*p*-Values
Soil properties				
pH	7.11 ± 0.09	7.73 ± 0.07	28.782	0.006
EC μS	1736.67 ± 777.63	357.33 ± 63.34	3.126	0.152
OM %	8.44 ± 1.17	4.99 ± 0.20	8.393	0.044
Ions (cations and anions)				
Na⁺ ppm	70.61 ± 28.87	58.42 ± 21.24	0.116	0.751
K⁺ ppm	334.65 ± 90.64	372.31 ± 79.34	0.098	0.770
Ca^2+^ ppm	0.77 ± 0.15	0.27 ± 0.04	10.843	0.030
Mg^2+^ ppm	0.49 ± 0.16	0.21 ± 0.03	2.770	0.171
SO_4_^2−^ ppm	49.25 ± 0.52	31.27 ± 0.61	511.249	0.001
NO_3_^−^ ppm	80.00 ± 11.55	73.33 ± 24.04	0.063	0.815
PO_4_^3−^ ppm	4.42 ± 2.04	2.67 ± 0.73	0.651	0.465
Ammonia (NH_4_^+^) ppm	133.33 ± 40.55	40.00 ± 11.55	4.900	0.091
Microelements				
Mn ppm	229.17 ± 182.30	269.58 ± 126.90	0.033	0.864
Zn mg/L	55.42 ± 11.59	45.00 ± 17.14	0.253	0.641
Fe ppm	10,564.58 ± 2597.91	16,856.25 ± 9276.55	0.427	0.549
Cr ppm	9.96 ± 6.17	12.97 ± 7.86	0.091	0.778
Co ppm	7.52 ± 2.82	9.76 ± 4.41	0.183	0.691
Cu ppm	10.21 ± 6.01	19.38 ± 9.57	0.658	0.463
Ni ppm	35.69 ± 25.95	66.25 ± 32.37	0.543	0.502

Data represented as means ± SE, *n* = 6.

**Table 3 plants-12-01231-t003:** Soil properties, ion concentrations, and microelement concentrations of soils in stands invaded by *Leucaena leucocephala* and adjacent stands with native vegetation in southwest Saudi Arabia.

	Invasive	Native	*t*-Test	*p*-Values
Soil properties				
pH	6.97 ± 0.22	7.64 ± 0.07	8.894	0.041
EC μS	7250.00 ± 4145.68	324.00 ± 12.22	2.791	0.170
OM %	12.22 ± 4.06	4.19 ± 0.09	3.901	0.119
Ions (cations and anions)				
Na⁺ ppm	297.22 ± 221.53	77.89 ± 31.77	0.960	0.383
K⁺ ppm	468.29 ± 41.84	536.73 ± 139.35	0.221	0.663
Ca^2+^ ppm	1.52 ± 1.07	0.38 ± 0.16	1.096	0.354
Mg^2+^ ppm	1.17 ± 0.44	0.48 ± 0.20	2.029	0.227
SO_4_^2−^ ppm	56.42 ± 12.45	41.07 ± 8.84	1.010	0.372
NO_3_^−^ ppm	173.33 ± 75.13	43.33 ± 8.82	2.953	0.161
PO_4_^3−^ ppm	2.33 ± 0.30	15.20 ± 10.33	1.551	0.281
Ammonia (NH_4_^+^) ppm	73.33 ± 13.33	53.33 ± 24.04	0.529	0.507
Microelements				
Mn ppm	73.33 ± 13.33	53.33 ± 24.04	0.529	0.507
Zn mg/L	57.71 ± 10.28	40.63 ± 15.66	0.831	0.413
Fe ppm	23,525.00 ± 9341.35	17,068.75 ± 10,154.34	0.219	0.664
Cr ppm	9.60 ± 1.21	12.24 ± 8.09	0.104	0.763
Co ppm	7.52 ± 0.34	8.09 ± 4.66	0.015	0.908
Cu ppm	12.29 ± 3.61	16.25 ± 10.23	0.133	0.734
Ni ppm	18.52 ± 2.13	45.50 ± 37.03	0.529	0.507

Data represented as means ± SE, *n* = 6.

**Table 4 plants-12-01231-t004:** Soil properties, ion concentrations, and microelement concentrations of soils in stands invaded by *Opuntia ficus-indica* and adjacent stands with native vegetation in southwest Saudi Arabia.

	Invasive	Native	*t*-Test	*p*-Values
Soil properties				
pH	7.08 ± 0.04	7.09 ± 0.03	0.084	0.786
EC μS	288.60 ± 68.97	191.23 ± 20.50	1.831	0.247
OM %	5.68 ± 2.14	3.36 ± 0.80	1.026	0.368
Ions (cations and anions)				
Na⁺ ppm	51.36 ± 6.46	35.95 ± 6.49	2.828	0.168
K⁺ ppm	545.34 ± 98.00	491.97 ± 167.64	0.076	0.797
Ca^2+^ ppm	0.27 ± 0.04	0.25 ± 0.00	0.143	0.725
Mg^2+^ ppm	0.20 ±0.02	0.17 ± 0.05	1.690	0.263
SO_4_^2−^ ppm	37.47 ± 0.96	31.50 ± 3.56	2.616	0.181
NO_3_^−^ ppm	126.67 ± 86.67	46.67 ± 6.67	0.847	0.409
PO_4_^3−^ ppm	17.70 ± 9.75	1.17 ± 0.92	2.848	0.167
Ammonia (NH_4_^+^) ppm	86.67 ± 13.33	73.33 ± 13.33	0.500	0.519
Microelements				
Mn ppm	312.50 ± 160.20	181.25 ± 76.38	0.547	0.501
Zn mg/L	81.25 ± 29.05	32.50 ± 6.68	2.674	0.177
Fe ppm	17,970.83 ± 5340.86	9552.08 ± 2189.11	2.127	0.218
Cr ppm	14.43 ± 3.84	33.13 ± 24.06	0.589	0.486
Co ppm	12.40 ± 0.74	10.10 ± 2.36	0.869	0.404
Cu ppm	18.96 ± 4.71	17.71 ± 4.65	0.036	0.859
Ni ppm	133.13 ± 17.51	52.78 ± 33.61	4.493	0.101

Data represented as means ± SE, *n* = 6.

## Data Availability

Not applicable.

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
