# Peer review of "Four Invasive Plant Species in Southwest Saudi Arabia Have Variable Effects on Soil Dynamics"

_plants, 2023, doi:10.3390/plants12061231_

Round 1

Reviewer 1 Report

This study presented the effects of four invasive plant species on soil properties using sampling and analyzing methods. The study provided some new knowledge considering the plant invasive events worldwide. The manuscript was generally well written, while several revisions are still needed before published. 

Except for all concerns highlighted in the attached file, I am highly worrying about the data analyzing method used in the study. If possible, please check if the data was able to explore using the nested ANOVA, which in my opinion, will provide more robust results, and enable the study to answer questions that the authors are trying to ask.

For other comments, please see attached.

Author Response

Plants-2219987, Reviewer 1 Comments:

This study presented the effects of four invasive plant species on soil properties using sampling and analyzing methods. The study provided some new knowledge considering the plant invasive events worldwide. The manuscript was generally well written, while several revisions are still needed before published. 

Response: Thank you.  We have endeavored to revise the manuscript as completely as possible, and we believe the revised manuscript has been greatly improved.

Except for all concerns highlighted in the attached file, I am highly worrying about the data analyzing method used in the study. If possible, please check if the data was able to explore using the nested ANOVA, which in my opinion, will provide more robust results, and enable the study to answer questions that the authors are trying to ask.

Response: While we appreciate this comment by Reviewer 1, we have chosen not to conduct the nested ANOVA analysis that was recommended. We believe our statistical analyses were appropriate for the data obtained in this study, but we have extensively edited the “Statistical analysis” part of the Material and Methods section of the revised manuscript. We used paired t-tests to determine differences between invaded and native soil samples for each of the four invasive species. We applied one-way ANOVA to test statistically significant differences in the soil properties, ion concentrations, and microelement concentrations for the stands of the four invasive plants and the native stands and Tukey’s honestly significant difference (HSD) test was used to determine the significance among means for treatments with a significant F-test.  Thus, we believe that the statistical analyses we conducted allowed us to analyze our data properly and draw reasonable inferences based on these analyses.

For other comments, please see attached.

Response: We have addressed almost every comment in the marked copy of the manuscript provided by Reviewer 1.

Reviewer 2 Report

Dear authors,

A review on the article «Four invasive plant species in southwest Saudi Arabia have variable effects on soil properties and soil microelements».
The article deals with the influence of four invasive species (Prosopis juliflora, Ipomoea carnea, Leucaena leucocephala and Opuntia ficus-indica). Eighteen parameters of soil properties and microelements were assessed. The results show that invasive species have different effects on the studied parameters and are not always aggressive on soil. The results are of undoubted practical and theoretical interest.
However, there are a number of comments to the paper:

1. The list of references needs to be updated. The paper clearly does not include enough works of the last few years.

2. In my opinion, more attention should be paid to the discussion of the reason for these different effects of invasive species on the soil.

3. It would be interesting to compare the results obtained in this article with the results of recently published article “Impacts of Nicotiana glauca Graham Invasion on the Vegetation Composition and Soil: A Case Study of Taif, Western Saudi Arabia".

Author Response

Plants-2219987, Reviewer 2 Comments

Dear authors,

A review on the article «Four invasive plant species in southwest Saudi Arabia have variable effects on soil properties and soil microelements».
The article deals with the influence of four invasive species (Prosopis juliflora, Ipomoea carnea, Leucaena leucocephala and Opuntia ficus-indica). Eighteen parameters of soil properties and microelements were assessed. The results show that invasive species have different effects on the studied parameters and are not always aggressive on soil. The results are of undoubted practical and theoretical interest.

Response: Thank you for this comment.

However, there are a number of comments to the paper:

  1. The list of references needs to be updated. The paper clearly does not include enough works of the last few years.

Response: Thank you for this suggestion.  In revising the manuscript, we now include more recent citations.  In addition, we also believe that based on the research and their findings, even older publications can be appropriate to cite and should be used.

  1. In my opinion, more attention should be paid to the discussion of the reason for these different effects of invasive species on the soil.

Response: We have now edited much of the revised manuscript (Abstract, Introduction, Discussion, and the Conclusion) to now focus on the idiosyncratic way plants and specifically invasive plants alter, or do not alter, soil properties and microelements.  The literature clearly points out that alteration of soil dynamics by invasive plants is species- and habitat-specific.  The revised manuscript now emphasizes these points.

  1. It would be interesting to compare the results obtained in this article with the results of recently published article “Impacts of Nicotiana glauca Graham Invasion on the Vegetation Composition and Soil: A Case Study of Taif, Western Saudi Arabia".

Response: Thank you for pointing out this paper.  We now cite it and Alfarhan et al. 2021 when describing invasive plant problems and research in southwest Saudi Arabia.  We have however not presented or discussed the results reported by Assaeed et al. 2021 because their soil analysis mostly assessed soil conditions (soil texture, soil moisture, cation content, etc.) associated with stands of vegetation invaded by Nicotiana glauca.  This study appear to have not assessed how N. glauca modified these soil conditions.  

Round 2

Reviewer 1 Report

The authors tried their best to revise the manuscript. 

It could be accepted once all ion revised to be "ion"(to be different from element) in the tables of the present version, which was also marked in the last round of review.

Author Response

Plants-2219987, Revised, Reviewer 1 Comments:

The authors tried their best to revise the manuscript. 

Response: Thank you.  As you indicate, we tried our bet to revise the manuscript according to most of your comments.  Indeed, we believe your comments and our revisions have resulted in a much improved manuscript.

It could be accepted once all ion revised to be "ion"(to be different from element) in the tables of the present version, which was also marked in the last round of review.

Response: Done.  We have revised Tables 1-4 by adding a new subsection labelled “Ions (cations and anions”.
